# Analysis of Cosmic Microwave Background with Deep Learning

**Siyu He** [*]
Department of Physics
Carnegie Mellon University
Pittsburgh, PA 15213, USA
siyuh@andrew.cmu.edu

**Siamak Ravanbakhsh**
Computer Science Department
University of British Columbia
Vancouver, BC V6T1Z4, Canada
siamakx@cs.ubc.ca

**Shirley Ho** [†‡]
Division of Physics
Lawrence Berkeley National Laboratory
Berkeley, CA 94720, USA
shirleyho@lbl.gov

## Abstract

The observation of Cosmic Microwave Background (CMB) has been one of the cornerstones in establishing the current understanding of the Universe. This valuable source of information consists of primary and secondary effects. While the primary source of information in CMB (as a Gaussian random field) can be efficiently analyzed using established statistical methods, CMB is also host to secondary sources of information that are more complex to analyze and understand. Here, we report encouraging preliminary results as well as some difficulties in using deep learning for prediction of the cosmological parameters and uncertainty estimates from the primary CMB. This opens the way to application of deep models in analysis of the secondary CMB and joint analysis of CMB with other modalities such as the large-scale structure

## 1 Introduction

The holy grail of field of cosmology is to understand the beginning, the evolution and content of the Universe. The observation of Cosmic Microwave Background (CMB) has been one of the cornerstones in establishing the current understanding of the Universe (Hinshaw et al., 2013; Planck Collaboration et al., 2015). Over the past decade, a standard cosmological model has emerged: with relatively few parameters, the model describes the evolution of the Universe and astronomical observations from a few to thousands of Megaparsecs. The observation and analysis of CMB maps offer a demanding test of this model. CMB maps are the fluctuation of temperature maps of the photons emitted from last scatter surface (38000 years after big bang). A demonstration of CMB maps are shown in Fig. 1. Since the primary CMB[1] is nearly purely Gaussian, we expect that all the information about the Universe from primary CMB can be fully encoded in the angular power-spectrum. Therefore, it is interesting to test if other novel machine learning methods such as deep learning can in principle extract all the information directly from the CMB maps, at least as much as a traditional angular power-spectrum analysis is capable of.

Here we present the first attempt at using advanced deep learning methods to predict cosmological parameters and their corresponding errors directly from the distribution of photons in Cosmic Microwave Background. Our objective is to show we could use deep models

---

[*]Division of Physics, Lawrence Berkeley National Laboratory, Berkeley, CA, United States

[†]Department of Physics, Carnegie Mellon University, Pittsburgh, PA, United States

[‡]Berkeley Center for Cosmological Physics, University of California, Berkeley, Berkeley, CA, United States

[1]The primary CMB is sourced by the initial fluctuations of the Universe, while the observed CMB also includes secondary fluctuations which are caused by the intervening matter impacting the photons which carry light from the early times at last scattering surface to us

to predict the parameters and uncertainty estimates for the predictions decently from primary CMB maps and we can move on to more complicated CMB maps with secondary anisotropy.

## 2 PRELIMINARY RESULTS

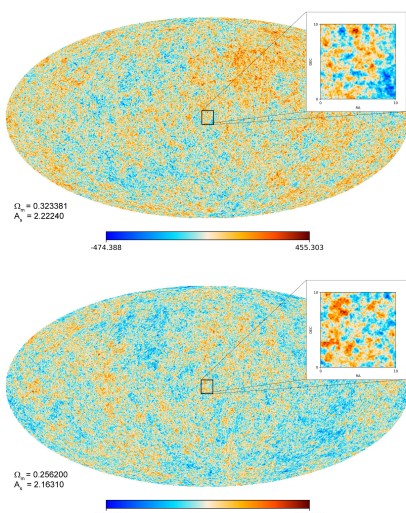

We use two models: 1) an 18-layers *deep residual network* (ResNet) (He et al., 2016) and; 2) *spherical convolutional neural network* (Spherical CNN Cohen et al., 2017), to predict matter density $\Omega_m$ and the amplitude of the primordial power spectrum $A_s$ from the CMB map. For each model, we generate 50,000 CMB maps using standard CMB package *healpy*[2] which takes CMB temperature angular power-spectra (generated from *CAMB*[3]) and generates Gaussian random fields in *Healpix*[4] formats. [5] The input datasets for the two models are different. In ResNet model, the input are 2-D images of CMB maps, each map covering 100 $deg^2$ of sky. We use *equatorial projection* to extract 10 deg $\times$10 deg maps around the equator and assume the small patch of the sky extracted is flat. In Spherical CNN model, the input are the whole sky images (around 41252.96 $deg^2$) on the sphere. Fig. 1 shows the demonstration of the CMB map input for both models.

Figure 1: An illustration of the input CMB maps. The map shown is the mollview projection of the entire sphere and the upper right small map shows the 100 $deg^2$ sky cut from the equator area. The small map can be viewed as flat. The two maps shown are CMB maps with different cosmological parameters.

The ResNet prediction of $(\Omega_m, A_s)$ shows an average relative error of (0.0078,0.0036),while for each $(\Omega_m, A_s)$, the ResNet predictions have a relatively small standard deviation of (0.0023,0.0063), indicating the ResNet is predicting quite accurately for individual images. For the Spherical CNN model, we are unable to make a decent prediction due to the fact that the CMB maps have a very high resolution and with current computation power we have, we need to aggressively pool to make the training computation manageable but which will smooth out the information in the CMB maps.

**PRODUCING UNCERTAINTY MEASURES.** To be comparable with existing physics-based analysis, we also need the uncertainty estimate for each predicted parameter. Thus, we include variance and co-variance predictions in the loss function on top of the ResNet model and maximize the log-likelihood. The mean relative error is (0.0033,0.0031) and the mean error prediction is (0.0012,0.0030). The result indicates that the model can successfully learn to quantify its own uncertainty at the time of prediction. An alternative approach to producing a measure of uncertainty is to use a *Bayesian Neural Network* (BNN Neal, 2012) and report the posterior predictive. However. our current results using Monte Carlo Dropout (Gal & Ghahramani, 2016) is not equally accurate. The mean relative error is (0.0035,0.022) for $(\Omega_m, A_s)$ while the predicted mean uncertainty is (0.0078,0.023). The predictions for $A_s$ are biased. We are investigating ways to improve the predictions using BNNs. In using MC Dropout we notice that the model has a difficulty in reducing its training error and its test performance is also significantly affected by the dropout. As an illustration of the result, Fig. 2 shows the mean-value predictions for $(\Omega_m, A_s)$ for 100 images with the same parameters but different random seeds using ResNet Model and Bayesian Neural Network model.

**STANDARD APPROACH.** We compare the performance of ResNet with a standard analysis of CMB maps in cosmology. For this, we produce maximum likelihood fit for both the angular power spectrum of the full maps and the 100 $deg^2$ maps. Fig. 3 shows a demonstration of the power spectra. The full map analysis shows a relative error of (0.0003,0.0002) for

---

[2]`https://github.com/healpy/healpy`
[3]`camb.info`
[4]`https://healpix.jpl.nasa.gov/`
[5]We use a 80%,10% and 10% split of the dataset for training, validation and test, respectively. We use data-augmentation using rotation and mirroring of the original maps.

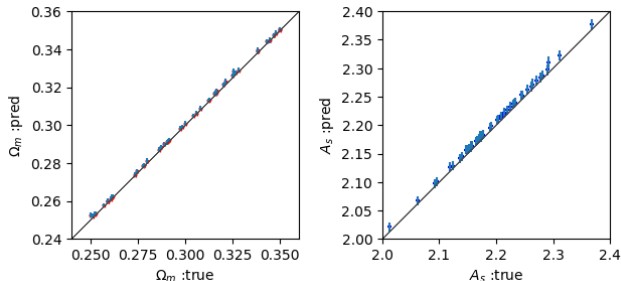

(a) $\Omega_m$ and $A_s$ predicted with error using ResNet model with - log(likehood) loss. Each prediction is an average of 100 maps.

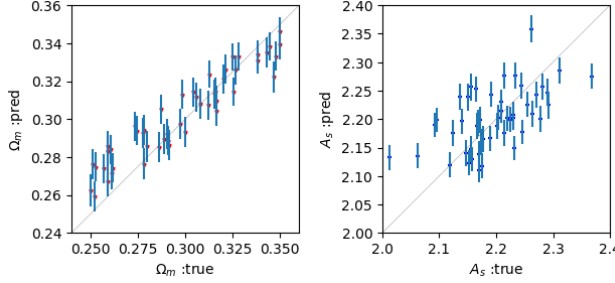

(b) $\Omega_m$ and $A_s$ predicted from Bayesian neural network with error prediction. Each prediction is an average of 100 maps.

Figure 2: $\Omega_m$ and $A_s$ predicted with with error

($\Omega_m$, $A_s$) while the analysis on the 100 $deg^2$ maps shows a relative error of (0.02, 0.08) (Note: We only have one map for analysis for each parameter ($\Omega_m$, $A_s$) and we are still in the process of calculating the maximum likelihood for 100 CMB maps for each set of parameters.).

The physics analysis is doing better than our method using the whole map. This is expected since the map are generated directly from the theory, where there is no loss of information and the noise is very small. However, for the small maps deep model might be doing a better job. This is because the small map size will induce a much more significant noise in the power spectra (note that This is not a fair comparison yet since we only have one realization of analysis for physics while 100 for deep learning. But from the one map comparison, it shows the deep learning method is doing a good job).

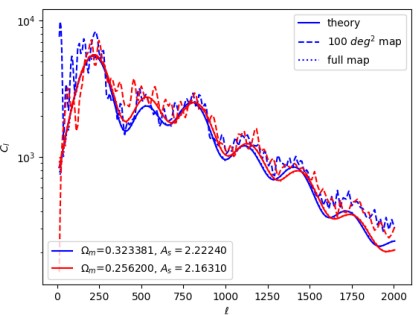

Figure 3: An illustration of the power spectra. As shown from the plot, the full map actually has small noise while the patches cut from the whole map has a large noise.

## 3   CONCLUSION AND FUTURE PLANS

In our work, we have tested the following two never tested hypotheses: a) the value of cosmological parameters from the Gaussian primary CMB maps can be predicted correctly by deep learning methods; b) the estimated error of each parameters can be predicted correctly by the deep learning methods. Though a full comparison with the physics analysis in under calculation, it's very promising to say the deep learning method will be doing as good as in physics analysis with the same type of input. Since primary CMB is a Gaussian random field, we can move on to more complicated CMB maps with secondary anisotropy. The perfect analysis of the power-spectrum does not extend secondary sources of information, where deep learning could be very effective. Furthermore, deep architectures allow us to easily combine the CMB with other cosmological dataset to do a joint analysis. We plan to explore these directions in the future.

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
