# OpenReview forum: "Analysis of Cosmic Microwave Background with Deep Learning"
_ICLR.cc/2018/Workshop — Accept_

### Official Review · AnonReviewer1 · 2018-02-26
**Preliminary work but an interesting direction**

**Rating:** 5
**Confidence:** 3

**Review:**

Imaging of the cosmic microwave background (CMB) provides some of the earliest information to constrain models of the early universe. Usually, these images are analysed using power-spectrum analysis (i.e. methods that use strong priors about the physical process) to, in this case, predict two parameters. This work shows that a ResNet trained on this task can predict these parameters (and proposes a spherical CNN but does not yet have results).

Pros:
- Paper is fairly well communicated.
- Using deep learning to improve analysis of physical observations is an exciting topic.

Cons:
- The work is very preliminary. All the predictions are on a simple task with simulated data, and for parameters which we have very good existing methods.
- While the domain is interesting, there is no novelty in the methods or insights applicable to other domains. ICLR attendees will not be surprised that ResNets were capable of solving what amounts to a simple image regression task.
- The proposal to extend these methods to more complex analysis where we do not have such strong priors is not fleshed out in any detail.

---

### Official Review · AnonReviewer3 · 2018-03-10
**Interesting application of deep learning**

**Rating:** 7
**Confidence:** 3

**Review:**

The paper presents an application of deep learning models for analyzing Cosmic Microwave Background maps. The authors use a ResNet and a Spherical CNN to estimate the matter density and amplitude of the power spectrum directly from the CMB maps. Additionally, the networks produce an estimation of the model's uncertainty as output, which turns to be more reliable compared to BNNs. Overall it is an interesting application of deep learning but the authors should make more clear what would be the advantage over the physics based methods (eg if they are perform equally well, why should someone use deep learning for such data).

---

### Decision · Program_Chairs · 2018-03-20
**ICLR 2018 Workshop Acceptance Decision**

**Decision:**

Accept

**Comment:**

Congratulations, your paper was accepted to the ICLR workshop.